# Only tails matter: Average-Case Universality and Robustness in the Convex Regime

## Abstract

Recent works have studied the average convergence properties of first-order optimization methods on distributions of quadratic problems. The average-case framework allows a more fine-grained and representative analysis of convergence than usual worst-case results, in exchange for a more precise hypothesis over the data generating process, namely assuming knowledge of the expected spectral distribution (e.s.d) of the random matrix associated with the problem. This work shows that a problem's asymptotic average complexity is determined by the concentration of eigenvalues near the edges of the e.s.d. We argue that having a priori information on this concentration is a more grounded assumption than complete knowledge of the e.s.d. basing our analysis on the approximate concentration is effectively a middle ground between the coarseness of the worst-case scenario convergence and the restrictive previous average-case analysis. We introduce the Generalized Chebyshev method, asymptotically optimal under a hypothesis on this concentration, and globally optimal when the e.s.d. follows a Beta distribution. We compare its performance to classical optimization algorithms, such as Gradient Descent or Nesterov's scheme, and we show that, asymptotically, Nesterov's method is universally nearly optimal in the average-case.

## 1 Introduction

The analysis of the average complexity of algorithms has a long story in computer science. Average-case complexity, for instance, drives much of the decisions made in cryptography (Bogdanov & Trevisan, 2006).

Despite their relevance, average-case analyses are difficult to extend to other algorithms, partly because of the intrinsic issue of defining a typical distribution over problem instances. Recently though, Pedregosa & Scieur (2020) derived a framework to systemically evaluate the complexity of first-order methods when applied on distributions of quadratic minimization problems. This is done by relating the average-case convergence rate to the *expected spectral distribution* (e.s.d) of the objective function's Hessian, which is a well-studied object on random matrix theory. Having access to this object in practice is a much stronger hypothesis when compared to the worst-case analysis that relies only on the values of the edges of this distribution.

Paquette et al. (2020) extended the average-case framework by introducing a noisy generative model for the problems. They further derived the average complexity of the Nesterov Accelerated Method (Nesterov, 2003) on a particular distribution. They showed the strong concentration of the metrics around a limiting value as dimensions go to infinity.

Scieur & Pedregosa (2020) showed that for a strongly convex problem with eigenvalues supported on a contiguous interval, the optimal average-case complexity converges asymptotically to the one given by the Polyak Heavy Ball method (Polyak, 1964) in the worst-case.

### 1.1 Current limitations of the average-case analysis

When analyzing the state of the art of average-case methods on quadratics problems, we observe significant limitations that we address in this paper. First, little is known

about the convergence rate on **convex problems**. Also, optimal average-case algorithms require an **exact estimation of the e.s.d** to guarantee an optimal convergence rate, their convergence rate under inexact e.s.d. is not known. Finally, the **non-smooth** is also discussed in (Pedregosa & Scieur, 2020), but with little details.

**Convex problems.** The minimization of non-strongly convex problems is drastically slower than their strongly convex counterpart, as Gradient Descent presents worst-case convergence in $\Theta(\frac{1}{t})$ and Nesterov is $\Theta(\frac{1}{t^2})$. In the strongly convex case, both the worst-case and average-case are asymptotically equal. However, little is known on optimal average-case rates for convex problems, as well as the average-case complexity of classical methods such as gradient descent or Nesterov's method, see (Paquette et al., 2020).

**Exact estimation of the e.s.d.** In (Pedregosa & Scieur, 2020), the theoretical study of optimal algorithms in the average-case requires an exact estimation of the e.s.d. of the problem class. Such estimation may be hard, nor impossible to obtain in practical scenarios. Despite showing good performance when the e.s.d. is estimated with empirical quantities, there are no theoretical guarantees on the performance of the method when the e.s.d. is poorly estimated. There is therefore a need to analyze the algorithm's performance under different notions of uncertainty on the spectrum. This allows a practitioner to choose the best algorithm for a practical problem, even with imperfect *a priori* information.

**Non-smooth.** Pedregosa & Scieur (2020) briefly introduce average-case optimal rates on non-smooth problems, when the e.s.d. is the Laguerre distribution $e^{-\lambda}$. In this paper, we extend the analysis to the generalized Laguerre distribution $\lambda^{\alpha} e^{-\lambda}$, $\alpha > -1$.

### 1.2 CONTRIBUTIONS

Our main contribution is a fine-grained analysis of the average-case complexity on convex quadratic problems: we show that a problem's complexity depends on the concentration of the eigenvalues of e.s.d. around the edges of their support. From this perspective, we propose a family of **optimal algorithms** in the average-case, analyze their **robustness**, and finally exhibit a **universality** result for Nesterov's method. More precisely,

- **(Optimal algorithms).** In Section 3, we propose the Generalized Chebyshev Method (GCM, Algorithm 1), a family of algorithms whose parameters depend on the concentration of the e.s.d. around the edges of their support. If the parameters of the GCM method are set properly, the algorithm converges at an optimal average-case rate (Theorem 3 for smooth problems, Theorem 6 for non-smooth problems), a rate that we show is faster than worst-case optimal methods like Nesterov acceleration. We show these rates to be representative of the practical performance of the algorithms in Fig. 6, and retrieve the classical worst-case rates as limits of the average-case (see Table 1).

- **(Robustness).** Developing an optimal algorithm requires the knowledge of the exact e.s.d. However, in practical scenarios, we only have access to an *approximation* of the e.s.d. In Theorem 2 in Section 4 we analyze the rate of GCM in the presence of such a mismatch. We also analyze the optimal average-case rates of distributions representing the smooth convex, non-smooth convex, and strongly convex settings and compare them with the worst-case rates (Table 1).

- **(Universality).** Finally, in Theorem 4, we analyze the asymptotic average-case convergence rate of Nesterov's method. We show that its convergence rate is nearly optimal (up to a logarithmic factor) under some natural assumptions over the data, namely a concentration of eigenvalues around 0 similar to the Marchenko-Pastur measure. This contributes to the theoretical understanding of the numerical efficiency of Nesterov's acceleration.

## 2 AVERAGE-CASE ANALYSIS

In this section, we recall the average-case analysis framework for random quadratic problems. The main result is Theorem 1, which relates the expected error to the *expected spectral*

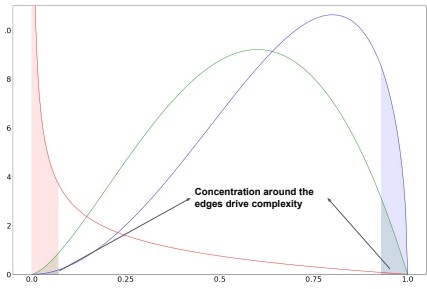

Figure 1: Representation of different spectra with different concentrations of eigenvalues around the edges of the support. The average-case rates for non-strongly problems are determined by these concentrations

| Regime | Worst-case | Average-Case |
|---|---|---|
| Strongly conv. | $(1 - \Theta(1/\sqrt{\kappa}))^t$ | $(1 - \Theta(1/\sqrt{\kappa}))^t$ |
| Smooth conv. | $1/t^2$ | $1/t^{2\xi+4}$ |
| Convex | $1/\sqrt{t}$ | $1/t^{\alpha+2}$ |

Table 1: Comparison between function value worst-case and average-case convergence. $\kappa$ is the condition number in the smooth strongly convex case. In the smooth convex case $\xi > -1$ is the concentration of eigenvalues around 0 (see Assumption 1) and in the non-smooth case we consider $d\mu \propto \lambda^\alpha e^{-\lambda}$

*distribution* and the *residual polynomial.* The one-to-one correspondence between the residual polynomials and first-order methods applied to quadratics will allow us to pose the problem of finding an optimal method as the best approximation problem in the space of polynomials.

We define a **random** quadratic problem:

**Problem 1.** *Let $\boldsymbol{H} \in \mathbb{R}^{d \times d}$ be a random symmetric positive-definite matrix independent to $\boldsymbol{x}^\star \in \mathbb{R}^d$, a random vector that is the solution to the problem. We define the random quadratic minimization problem as*

$$\min_{\boldsymbol{x} \in \mathbb{R}^d} \left\{ f(\boldsymbol{x}) := \frac{1}{2}(\boldsymbol{x} - \boldsymbol{x}^\star)^\top \boldsymbol{H}(\boldsymbol{x} - \boldsymbol{x}^\star) \right\}. \tag{OPT}$$

*We are interested on minimizing the expected errors $\mathbb{E}\|f(\boldsymbol{x}_t) - f(\boldsymbol{x}^\star)\|$, the expected function-value gap, and $\mathbb{E}\|\nabla f(\boldsymbol{x}_t)\|^2$, the expected gradient norm, where $\boldsymbol{x}_t$ is the $t$-th update of a first-order method starting from $\boldsymbol{x}_0$ and $\mathbb{E}$ is the expectation over the random variables $\boldsymbol{H}, \boldsymbol{x}_0$ and $\boldsymbol{x}^\star$.*

The expectation we consider is over the problem and not over any randomness of the algorithm.

In this paper, we consider the class of *first-order methods* (F.O.M's) to minimize (OPT). Methods in this class construct the iterates $\boldsymbol{x}_t$ as

$$\boldsymbol{x}_t \in \boldsymbol{x}_0 + \mathbf{span}\{\nabla f(\boldsymbol{x}_0), \dots, \nabla f(\boldsymbol{x}_{t-1})\}. \tag{1}$$

That is, $\boldsymbol{x}_t$ belongs to the span of previous gradients. This class of algorithms includes for instance gradient descent and momentum, but not quasi-Newton methods since the preconditioner could allow the iterates to go outside of the span. Furthermore, we will only consider *oblivious* methods, that is, methods in which the coefficients of the update are known in advance and do not depend on previous updates. This leaves out some methods such as conjugate gradient or methods with line-search.

**From First-Order Method to Polynomials.** There is an intimate link between first-order methods and polynomials that simplifies the analysis of quadratic objectives. The next proposition shows that, with this link, we can assign to each optimization method a polynomial that determines its convergence. Following Fischer (1996), we will say a polynomial $P_t$ is *residual* if $P_t(0) = 1$.

**Proposition 1.** *(Hestenes et al., 1952) Let $\boldsymbol{x}_t$ be generated by a first-order method. Then there exists a residual polynomial $P_t$ of degree $t$, that verifies*

$$\boldsymbol{x}_t - \boldsymbol{x}^\star = P_t(\boldsymbol{H})(\boldsymbol{x}_0 - \boldsymbol{x}^\star). \tag{2}$$

**Remark 1.** *If the first-order method is further a **momentum method**, i.e.*

$$\boldsymbol{x}_{t+1} = \boldsymbol{x}_t + h_t \nabla f(\boldsymbol{x}_t) + m_t(\boldsymbol{x}_t - \boldsymbol{x}_{t-1}).$$

*We can determine the polynomials by the recurrence $P_0 = 1$ and*

$$P_{t+1}(\lambda) = P_t(\lambda) + h_t \lambda P_t(\lambda) + m_t(P_t(\lambda) - P_{t-1}(\lambda)).$$

*We note that while most popular F.O.M's can be posed as a momentum method, the Nesterov method cannot.*

A convenient way to collect statistics on the spectrum of a matrix is through its *empirical spectral distribution*.

**Definition 1** (Expected spectral distribution (e.s.d)). *. Let $\boldsymbol{H}$ be a random matrix with eigenvalues $\{\lambda_1, \ldots, \lambda_d\}$. The **empirical spectral distribution** of $\boldsymbol{H}$, called $\mu_{\boldsymbol{H}}$, is the probability measure*

$$\mu_{\boldsymbol{H}} := \frac{1}{d}\sum_{i=1}^{d}\delta_{\lambda_i}, \tag{3}$$

*where $\delta_{\lambda_i}$ is the Dirac delta, a distribution equal to zero everywhere except at $\lambda_i$ and whose integral over the entire real line is equal to one.*

*Since $\boldsymbol{H}$ is random, the empirical spectral distribution $\mu_{\boldsymbol{H}}$ is a random variable in the space of measures. Its expectation over $\boldsymbol{H}$ is called the **expected spectral distribution** and we denote it*

$$\mu := \mathbb{E}_{\boldsymbol{H}}[\mu_{\boldsymbol{H}}]. \tag{4}$$

We can link the e.s.d. of $\boldsymbol{H}$ to the convergence of a first-order method on the distribution of $\boldsymbol{H}$. In the following we will consider $\boldsymbol{x}_0 - \boldsymbol{x}^\star$ and $\boldsymbol{H}$ to be independent, with $\boldsymbol{x}_0 - \boldsymbol{x}^\star$ sampled isotropically.

**Theorem 1.** *Let $\boldsymbol{x}_t$ be generated by a first-order method associated to the polynomial $P_t$, the measure $\mu$ the e.s.d. of $H$, and $\mathbb{E}[(\boldsymbol{x}_0 - \boldsymbol{x}^\star)(\boldsymbol{x}_0 - \boldsymbol{x}^\star)^T] = R^2 \boldsymbol{I}$ for some constant $R$. Then we can write the convergence metrics at time step $t$ as*

$$\mathbb{E}[\|\boldsymbol{x}_t - \boldsymbol{x}^\star\|^2] = R^2 \int P_t^2(\lambda)d\mu(\lambda), \qquad \mathbb{E}[f(\boldsymbol{x}_t) - f(\boldsymbol{x}^\star)] = R^2 \int P_t^2(\lambda)\lambda d\mu(\lambda)$$

$$and \qquad \mathbb{E}[\|\nabla f(\boldsymbol{x}_t)\|_2^2] = R^2 \int P_t^2(\lambda)\lambda^2 d\mu(\lambda). \tag{5}$$

This shows that polynomials are a powerful abstraction as they allow us to write all of our convergence metrics within the same framework . For simplicity, we set $R^2 = 1$ and we will refer directly to the polynomials associated to a given method. We will refer to objective $l$ as the one associated to the added $\lambda^l$ term, i.e. the function-value is objective $l = 1$.

This framework is linked to the field of **orthogonal polynomials** by the next proposition. We construct an optimal method w.r.t. a given distribution through a family of orthogonal polynomials associated to it.

**Proposition 2** ((Pedregosa & Scieur, 2020)). *Let $P_t^l$ be defined as*

$$P_t^l := \arg\min_{P_t(0)=1} \int P_t^2(\lambda)\lambda^l d\nu(\lambda). \tag{6}$$

*Then $(P_t^l)$ is the family of residual orthogonal polynomials w.r.t. to $\lambda^{l+1}d\nu$.*

This theorem further implies that the optimal first-order method is a momentum method as Favard's theorem Marcellán & Álvarez-Nodarse (2001) tells us the orthogonal polynomials w.r.t. a given distribution are related through a **three term recurrence**,

$$P_{t+1}(\lambda) = a_t P_t(\lambda) + b_t \lambda P_t(\lambda) + (1 - a_t)P_{t-1}(\lambda). \tag{7}$$

Following Remark 1, the optimal method is derived from this recurrence as

$$\boldsymbol{x}_{t+1} = \boldsymbol{x}_t + (a_t - 1)(\boldsymbol{x}_t - \boldsymbol{x}_{t-1}) + b_t \nabla f(\boldsymbol{x}_t). \tag{8}$$

## 3 METHODS

Being able to write the rates in terms of the *expected spectral distribution* ties the average-case framework to the field of *random matrix theory*. Indeed, because of results from this field, certain e.s.d's are considered more natural than others. Indeed, it can be shown that the same distribution arises when we take the gram matrix of random centered i.i.d. features with variance $\sigma^2$: the **Marchenko Pastur** distribution.

**Definition 2** (MP distribution). *The Marchenko Pastur distribution associated with the parameter $r$ and with scale $\sigma^2$ is given by*

$$d\mu_{MP}(\lambda) = \frac{1}{2\pi\sigma^2}\frac{\sqrt{(\lambda^+ - \lambda)(\lambda - \lambda^-)}}{r\lambda}, \tag{9}$$

*with $\lambda^+ = \sigma^2(1 + \sqrt{r})^2$, $\lambda^- = \sigma^2 \max(0, (1 - \sqrt{r})^2)$.*

The Marchenko Pastur distribution $\mu_{MP}$ can be considered a natural first model for e.s.d's as it arises universally from matrices with i.i.d. entries,under mild low moment assumptions, there is no specific distribution of the matrix to be considered. It can be seen as a model for the white-noise in the data. When $r = 1$, i.e. $n = d$, we have $d\mu_{MP} \propto \lambda^{-1/2}\sqrt{\lambda^+ - \lambda}$.

Pedregosa & Scieur (2020) first derived the optimal method w.r.t. $\mu_{MP}$, and Paquette et al. (2020) derived Nesterov's rates under the distribution. As we are concerned with being robust, a natural step is to consider the Beta weights.

**Definition 3.** *The (generalized) Beta weights with parameters $\tau, \xi$ and scale $L$ are given by the (non-normalized) pdf*

$$d\mu(\lambda) = \lambda^\xi(L - \lambda)^\tau. \tag{10}$$

This family of distribution generalizes the MP distribution, and both have similar concentrations near 0 when $\xi \approx -1/2$.

The optimal method w.r.t. $\mu$ and objective $l$ is associated to a shifted Jacobi polynomial $\tilde{P}_t^{\alpha,\beta}$ with $\beta = \xi + l + 1, \alpha = \tau$. When $\alpha = \beta = -1/2$, we retrieve the *Chebyshev Method* (Flanders & Shortley, 1950). As such, we name our proposed methods the *Generalized Chebyshev Method* (GCM).

---

**Algorithm 1:** GCM$(\alpha, \beta)$

---

**Inputs**: Initial vector $\boldsymbol{x}_0$, function $f$, smoothness parameter estimate $L$

$\boldsymbol{x}_{-1} \leftarrow \boldsymbol{0}, \delta_0 \leftarrow 0$

**for** $t = 1, \ldots, T$ **do**

$\quad a_t \leftarrow -\frac{2\left(\beta^2 + \alpha\beta + (2t+1)(\alpha+\beta) + 2t^2 + 2\right)(2t+\alpha+\beta+1)}{2(t+1)(t+\alpha+\beta+1)(2t+\alpha+\beta)}$

$\quad b_t \leftarrow \frac{(2t+\alpha+\beta+1)(2t+\alpha+\beta+2)}{L(t+1)(t+\alpha+\beta+1)}$

$\quad \gamma_t \leftarrow -\frac{(t+\alpha)(t+\beta)(2t+\alpha+\beta+2)}{(t+1)(t+\alpha+\beta+1)(2t+\alpha+\beta)}$

$\quad \delta_t \leftarrow \frac{1}{a_t + \gamma_t \delta_{t-1}}$

$\quad \boldsymbol{x}_t \leftarrow \boldsymbol{x}_{t-1} + (\delta_t a_t - 1)(\boldsymbol{x}_{t-1} - \boldsymbol{x}_{t-2}) + \delta_t b_t \nabla f(\boldsymbol{x}_{t-1})$

---

We'll consider the Nesterov's method used in Paquette et al. (2020), which is defined by the iterations:

$$\boldsymbol{x}_{t+1} = \boldsymbol{y}_t - \frac{1}{L}\nabla f(\boldsymbol{y}_t) \tag{11}$$

$$\boldsymbol{y}_{t+1} = \boldsymbol{x}_{t+1} + \frac{t}{t+3}(\boldsymbol{x}_{t+1} - \boldsymbol{x}_t) \tag{12}$$

We also consider the Laguerre method, which is optimal w.r.t. $d\mu(x) = \frac{x^\alpha e^{-x}}{\Gamma(\alpha+1)}$, taking $\alpha$ as a parameter. This method is proposed to optimize non-smooth functions.
Both these methods are generalizations of ones that have been proposed in Pedregosa &

Scieur (2020). We show that Algorithm 1 corresponded to polynomials $\tilde{P}_t^{\alpha,\beta}$ and derive the Laguerre method in appendix B.

**Remark 2.** *The Generalized Chebyshev takes the largest eigenvalue $L$ as a parameter, but the rates we will show are robust to an overestimation of $L$.*

## 4 ROBUST AVERAGE-CASE RATES

We will state our assumption over the spectral distributions. It effectively allows us to parametrize all of our distributions of interest in a way that characterizes the asymptotic convergence, diving them into equivalence classes.

**Assumption 1.** *We will write $\nu_{\tau,\xi}$ for a continuous distribution supported in $(0, L]$ s.t. $\nu'_{\tau,\xi}(x) > 0$ for $x \in [0, L]$, $d\nu_{\tau,\xi} = \Theta(\lambda^\xi)$ near 0 and $d\nu_{\tau,\xi} = \Theta((L - \lambda)^\tau)$ near $L$.*

Assumption 1 is quite nonrestrictive, in that, the spectral distribution of the Hessian for any smooth convex problem can be identified with some $\tau, \xi$ in this class. It is a milder assumption than (1) assuming complete knowledge of the spectrum of the Hessian or (2) the specific distribution on the entries of your data. Moreover the assumption encompasses the frequently used MP (e.g. Martin & Mahoney (2021); Pennington & Bahri (2017)) and Uniform distributions We note there's no need to consider eigenvalues situated at 0 as they do not contribute to the optimization process.

The $\xi$ works as a measure of how close we are to the worst-case scenario, as it approaches $-1$. Samples in finite dimension of distributions with high values of $\xi$ will work as strongly convex functions in practice.

We show that $\nu_{\tau,\xi}$ indeed behaves like an equivalence class when considering the asymptotics of the convergence of the methods: only the concentrations near the edge matter. We do this by singling out from each of these classes the beta distributions for which we can compute the rates, then show the rates to be the same inside $\nu_{\tau,\xi}$.

**Theorem 2** (GCM average-case rates). *A Generalized Chebyshev Method with parameters $(\alpha, \beta)$ applied to a problem with e.s.d. as in Assumption 1 has average-case rates*

$$\mathbb{E}[f(\boldsymbol{x}_t) - f(\boldsymbol{x}^\star)] \sim L \cdot C_{1,\nu}^{\alpha,\beta} \begin{cases} t^{-1-2\beta} & \text{if } \alpha < \tau + 1/2 \text{ and } \beta < \xi + 3/2 \\ t^{-2(\xi+2)} \log t & \text{if } \alpha = \tau + 1/2 \text{ and } \beta = \xi + 3/2 \\ t^{2(\max\{\alpha-\beta-\tau, -\xi-1\}-1)} & \text{if } \alpha > \tau + 1/2 \text{ or } \beta > \xi + 3/2 \end{cases},$$
(13)

$$\mathbb{E}[||\nabla f(\boldsymbol{x}_t)||_2^2] \sim L^2 \cdot C_{2,\nu}^{\alpha,\beta} \begin{cases} t^{-1-2\beta} & \text{if } \alpha < \tau + 1/2 \text{ and } \beta < \xi + 5/2 \\ t^{-2(\xi+3)} \log t & \text{if } \alpha = \tau + 1/2 \text{ and } \beta = \xi + 5/2 \\ t^{2(\max\{\alpha-\beta-\tau, -\xi-2\}-1)} & \text{if } \alpha > \tau + 1/2 \text{ or } \beta > \xi + 5/2 \end{cases},$$
(14)

*where $C_\nu^{\alpha,\beta}$ is a distribution dependent constant.*

Theorem 2, which is illustrated by fig. 2 shows that overestimating $\beta$, and underestimating $\alpha$ will still leave us with the optimal asymptotic rates, so a good rule of thumb for calibrating the algorithm is to use high $\beta$ and low $\alpha$.

Theorem 3 shows that a proper choice of $\alpha, \beta$ can indeed make the Jacobi polynomial asymptotically optimal w.r.t. to any $\nu_{\tau,\xi}$.

**Theorem 3** (Optimal Rates). *Let $\nu$ follow Assumption 1. The optimal asymptotic average-case rates for $\mathbb{E}[f(\boldsymbol{x}_t) - f(\boldsymbol{x}^\star)]$ and $\mathbb{E}[||\nabla f(\boldsymbol{x}_t)||_2^2]$ are attained by the GCM with parameters $(\tau, \xi + 2)$ and $(\tau, \xi + 3)$, respectively, and read*

$$\mathbb{E}[f(\boldsymbol{x}_t) - f(\boldsymbol{x}^\star)] = \Theta(t^{-2(\xi+2)}), \qquad \mathbb{E}[||\nabla f(\boldsymbol{x}_t)||_2^2] = \Theta(t^{-2(\xi+3)}).$$

For the function value ($l = 1$), we find rates that approach $t^{-2}$ as $\xi \to -1$, showing the worst-case as a limit (over the considered distribution) on the average-case.

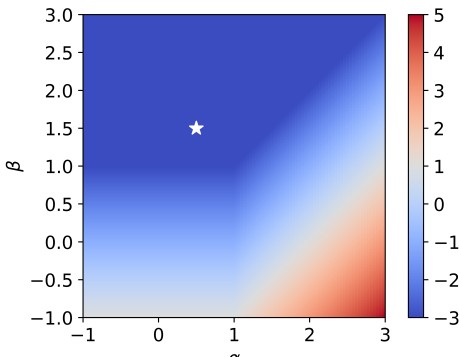

| Method | Parameters $(\tau, \xi)$ | |
|---|---|---|
| | $(\frac{1}{2}, \frac{1}{2})$ | $(\frac{1}{2}, -\frac{1}{2})$ |
| GCM($\alpha = \frac{1}{2}, \beta = \frac{5}{2}$) | $t^{-5}$ | $t^{-3}$ |
| GCM($\alpha = \frac{1}{2}, \beta = \frac{3}{2}$) | $t^{-4}$ | $t^{-3}$ |
| Nesterov | $t^{-4}$ | $t^{-3} \log t$ |
| Gradient Descent | $t^{\frac{-5}{2}}$ | $t^{\frac{-3}{2}}$ |

Figure 3 & Table 1: The figure illustrates the robustness of the Generalized Chebyshev Method with parameters $(\alpha, \beta)$ for a *fixed problem* corresponding to the Marchenko-Pastur distribution $(\tau = \frac{1}{2}, \xi = -\frac{1}{2})$. The color represents the exponent $a$ of the average-case rate $O(t^a)$ of the method for different values of $\alpha$ and $\beta$. The white star represents the optimal tuning and the blue area is the set of parameters for which the method converges. Note we have a large of region that guarantee the same optimal asymptotic rate. The table compares the asymptotic average-case rates for the function-value for different methods with different $(\tau, \xi)$ values.

We remark that the above theorems imply that, at least asymptotically, the GCM is robust for a suboptimal choice of parameter $\beta$ up to $1/2$ below the optimal choice and infinitely above.

For completeness, we also derive worst-case rates for the GCM:

**Proposition 3** (GCM worst-case rates). *Let $f$ be a convex, $L$-smooth quadratic function. Then, for the Generalized Chebyshev Method with parameters $(\alpha, \beta)$, we have worst-case rates*

$$f(\boldsymbol{x}_t) - f(\boldsymbol{x}^\star) \leq C_1 L \begin{cases} t^{2(\alpha-\beta)} & if \quad \alpha > \beta - 1 \\ t^{-1-2\beta}, & if \quad \alpha \leq \beta - 1 \quad \beta \leq \frac{1}{2} \\ t^{-2}, & if \quad \alpha \leq \beta - 1 \quad \beta \geq \frac{1}{2} \end{cases}, \tag{15}$$

$$||\nabla f(\boldsymbol{x}_t) - f(\boldsymbol{x}^\star)|| \leq C_2 L^2 \begin{cases} t^{2(\alpha-\beta)} & if \quad \alpha > \beta - 2 \\ t^{-1-2\beta}, & if \quad \alpha \leq \beta - 2 \quad \beta \leq 3/2 \\ t^{-4}, & if \quad \alpha \leq \beta - 2 \quad \beta \geq 3/2 \end{cases}. \tag{16}$$

For a reasonable choice of $\alpha, \beta$, i.e. $\beta \geq \frac{1}{2}$, $\alpha \leq \beta - 1$. the function value achieves the theoretical lower bound of $t^{-2}$.

We now analyze the convergence of the Nesterov method. Nesterov (2003) has shown that it matches up to a constant factor a lower bound on the worst-case complexity of non strongly convex problems. A natural question is if this performance would translate to good average-case rates. To do so, we will extend Paquette et al. (2020) proof for the Nesterov method rates under the MP distribution.

**Theorem 4** (Nesterov average-case rates). *Let $\nu$ as in Assumption 1. Then for the Nesterov method, we have average-case rates*

$$\mathbb{E}[f(\boldsymbol{x}_t) - f(\boldsymbol{x}^\star)] \sim C'_{1,\nu} \begin{cases} t^{-2(\xi+2)} & if\ \xi < -1/2 \\ t^{-3} \log t & if\ \xi = -1/2 \\ t^{-(\xi+7/2)} & if\ \xi > -1/2 \end{cases}, \quad \mathbb{E}[||\nabla f(\boldsymbol{x}_t)||_2^2] \sim C'_{2,\nu} t^{-(\xi+9/2)}. \tag{17}$$

The difference between the asymptotic average-case rates of Nesterov and the optimal ones are $t^{\xi+l-1/2}$, when $\xi + l > 1/2$, $\log t$ when $\xi + l = 1/2$ and $0$ otherwise. This shows that Nesterov is almost optimal when the concentrations near $0$ are relatively high, i.e. low $\xi$.

**Theorem 5** (Gradient Descent average-case rates). *Let $\nu$ as in Assumption 1. Then for gradient descent*

$$\mathbb{E}[f(\boldsymbol{x}_t) - f(\boldsymbol{x}^\star)] = \Theta(t^{-(\xi+2)}), \qquad \mathbb{E}[||\nabla f(\boldsymbol{x}_t)||_2^2] = \Theta(t^{-(\xi+3)}). \tag{18}$$

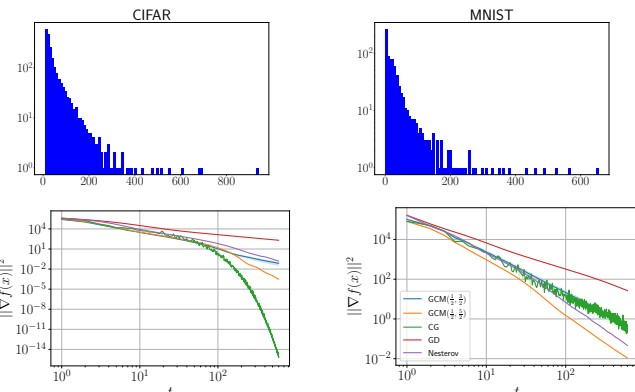

Figure 4: *Above:* Empirical spectrum for the covariance matrix of the features. *Below:* Gradient norms throughout iterations. *Left:* CIFAR-10 Inception features *Right*: MNIST features. Here we choose to compare gradient norms as the minimum function value is not known. The properly tuned GCM achieves remarkable performance under these non-synthetic spectrum's.

Observe for the function value that the rate for Nesterov is $t^{-2}$ and the rate for Gradient Descent is $t^{-1}$ when $\xi \to -1$.

Lastly, we consider the optimal rates for a Gamma distribution.

**Theorem 6** (Laguerre method rates). *Let $\alpha > -1$ and $\mu_\alpha$, $\alpha > -1$ be a Gamma distribution, i.e. $d\mu_\alpha(x) = \frac{x^\alpha e^{-x}}{\Gamma(\alpha+1)}$. The optimal rates are given by the Laguerre method of appropriate tuning and*

$$\mathbb{E}[f(\boldsymbol{x}_t) - f(\boldsymbol{x}^\star)] = \Theta(t^{-(\alpha+2)}). \tag{19}$$

Note that this result does not have the same universality as the others because of the non-compacity of the distribution's support. These rates are contrasted to the worst-case lower bound on the optimization of non-smooth functions by first-order methods, which gives

$$f(\boldsymbol{x}_k) - f(\boldsymbol{x}^\star) \geq \frac{C}{\sqrt{t}}.$$

These rates are not found when $\alpha \to -1$, indicating that the worst-case is especially pessimistic in this scenario.

**Remark 3.** *All of the expected rates we state are almost deterministic on the high dimensional setting as per the concentration results shown in* Paquette et al. (2020)

## 5 EXPERIMENTS

We simulate the e.s.d's in two ways. The Marchenko Pastur distribution, which we sample by taking $\boldsymbol{H} = \boldsymbol{X}\boldsymbol{X}^T$ where $\boldsymbol{X}$ has i.i.d. gaussian samples. This enables us to simulate $(\tau, \xi)$ values of $(1/2, -1/2)$. Other values of $(\tau, \xi)$ are simulated by sampling $\Lambda \in \mathbb{R}^d$ from the corresponding Beta distribution and taking $\boldsymbol{H} = \boldsymbol{U}\,\mathbf{diag}(\Lambda)\boldsymbol{U}^T$, where $\boldsymbol{U}$ is an independently sampled orthonormal matrix.

We let $\boldsymbol{x}^\star = 0$ and sample $\boldsymbol{x}_0$ from a centered Gaussian distribution, the dynamics are the same as in the general case. In all experiments we use the problem's instance largest eigenvalue to calibrate each method (e.g. Gradient Descent's stepsize is $1/L$). Our theoretical rates in Theorem 5 and Theorem 4 respectively for the Nesterov method and Gradient Descent are precise under the approximate range $-1 < \xi < 0$ as we show in Figure 6. Distributions with higher $\xi$ need many samples otherwise they behave as strongly convex functions.

The same is not true for the Generalized Chebyshev Method. If $\beta < \beta^\star$ or $\xi$ is low the empirical findings diverge from the theoretical. We believe this is due to numerical instability

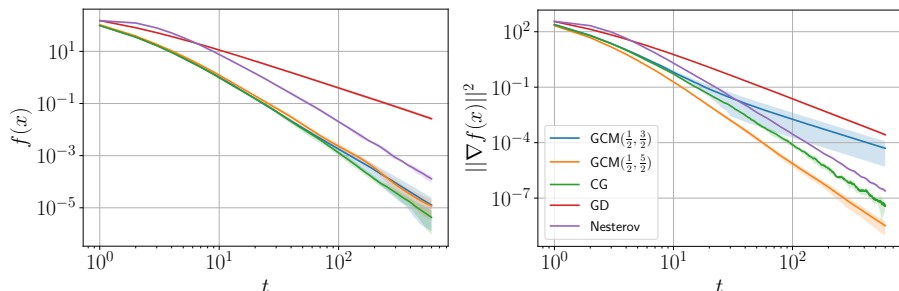

Figure 5: Rates for a synthetic problem, simulating the Marchenko Pastur distribution. Note that both tunings of the GCM achieve performance in function value very close to the one of Conjugate Gradient, which is optimal for every draw of the problem.

under these regimes as the metrics also have much larger variance than in the other regimes. We've not been able though to pinpoint the exact source of this supposed instability. This is shown in appendix D.

The GCM with $\beta > \beta^\star$ performs corresponding to the theory, and it's non-asymptotically very close to the performance of $\beta^\star$. High values of $\beta$ also perform very well on non-synthetic data, suggesting in practice we should use these values.

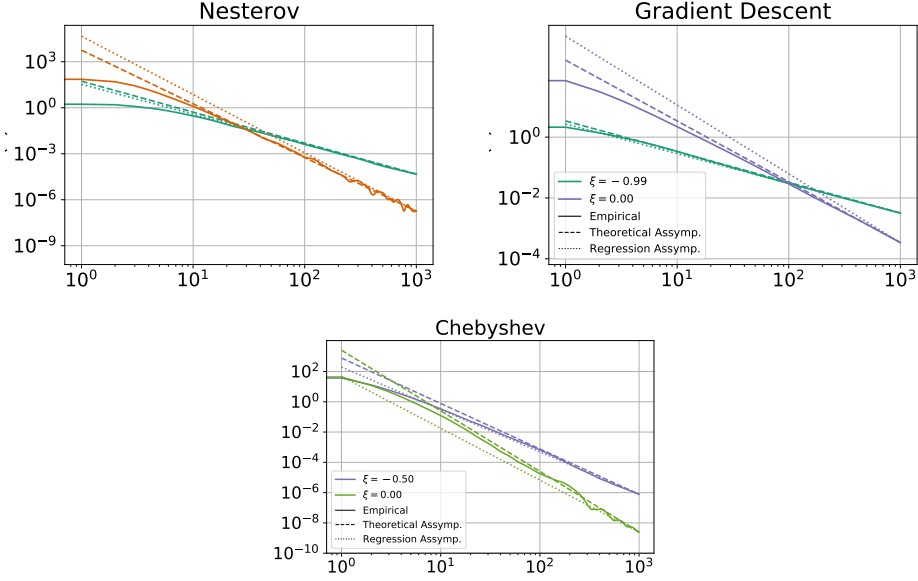

Figure 6: Comparison between experiments run on synthetic Beta distribution and theoretical asymptotic. Y-axis is the function value

## 6 CONCLUSION AND FURTHER WORK

In this paper, we've established that the assymptotic convergence of first order methods on quadratic problems in the convex regime depend on the concentration of the Hessian's eigenvalue near the edges of the spectrum's support. We further contributed to the theoretic understanding of the Nesterov's method performance and established the contrast between the worst-case and average-case in the main regimes considered in Optimization.

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

## A  PROOFS OF SECTION 2

**Theorem 1.** *Let $\boldsymbol{x}_t$ be generated by a first-order method associated to the polynomial $P_t$, the measure $\mu$ the e.s.d. of $H$, and $\mathbb{E}[(\boldsymbol{x}_0 - \boldsymbol{x}^\star)(\boldsymbol{x}_0 - \boldsymbol{x}^\star)^T] = R^2\boldsymbol{I}$ for some constant $R$. Then we can write the convergence metrics at time step $t$ as*

$$\mathbb{E}[\|\boldsymbol{x}_t - \boldsymbol{x}^\star\|^2] = R^2 \int P_t^2(\lambda)d\mu(\lambda), \qquad \mathbb{E}[f(\boldsymbol{x}_t) - f(\boldsymbol{x}^\star)] = R^2 \int P_t^2(\lambda)\lambda d\mu(\lambda)$$

$$and \qquad \mathbb{E}[\|\nabla f(\boldsymbol{x}_t)\|_2^2] = R^2 \int P_t^2(\lambda)\lambda^2 d\mu(\lambda). \tag{5}$$

*Proof.* We remark that by the definition of the expected spectral distribution $\mu$ of $\boldsymbol{H}$, we have for continuous $g$

$$\mathbb{E}_H[g(tr(\boldsymbol{H}))] = \int g(\lambda)\,d\mu(\lambda) \tag{20}$$

We know that $\boldsymbol{x}_t - \boldsymbol{x}^\star = P_t(\boldsymbol{H})(\boldsymbol{x}_0 - \boldsymbol{x}^\star)$. We can write $\|\boldsymbol{x}_t - \boldsymbol{x}^\star\|^2$ in terms of a trace and use the independence of $\boldsymbol{H}$ and $\boldsymbol{x}_0 - \boldsymbol{x}^\star$ to connect it to the e.s.d.:

$$\mathbb{E}\|\boldsymbol{x}_t - \boldsymbol{x}^\star\|^2 = \mathbb{E}[tr((\boldsymbol{x}_0 - \boldsymbol{x}^\star)^T P_t(\boldsymbol{H})^2(\boldsymbol{x}_0 - \boldsymbol{x}^\star))] \tag{21}$$

$$= \mathbb{E}_{\boldsymbol{H},\boldsymbol{x}_0 - \boldsymbol{x}^\star}[tr(P_t(\boldsymbol{H})^2(\boldsymbol{x}_0 - \boldsymbol{x}^\star)(\boldsymbol{x}_0 - \boldsymbol{x}^\star)^T]] \tag{22}$$

$$= \mathbb{E}_{\boldsymbol{H}}\left[P_t(\boldsymbol{H})^2\mathbb{E}_{\boldsymbol{x}_0 - \boldsymbol{x}^\star}[(\boldsymbol{x}_0 - \boldsymbol{x}^\star)(\boldsymbol{x}_0 - \boldsymbol{x}^\star)^T])\right] \tag{23}$$

$$= R^2\mathbb{E}_{\boldsymbol{H}}[P_t(tr(\boldsymbol{H}))^2] = R^2 \int P_t(\lambda)^2\,d\mu(\lambda) \tag{24}$$

For the gradient and function value the reasoning is the same by noticing that

$$\mathbb{E}[f(\boldsymbol{x}_t) - f(\boldsymbol{x}^\star)] = \mathbb{E}[tr((\boldsymbol{x}_0 - \boldsymbol{x}^\star)^T P_t(\boldsymbol{H})\boldsymbol{H}P_t(\boldsymbol{H})(\boldsymbol{x}_0 - \boldsymbol{x}^\star))] \tag{25}$$

$$= \mathbb{E}_{\boldsymbol{H}}[(\lambda P_t)(tr(H))^2], \tag{26}$$

where $\lambda P_t$ is also a polynomial. As $\nabla f(\boldsymbol{x}_t) = \boldsymbol{H}(\boldsymbol{x}_t - \boldsymbol{x}^\star)$.

$$\mathbb{E}\|\nabla f(\boldsymbol{x}_t))\|^2 = \mathbb{E}[tr((\boldsymbol{x}_0 - \boldsymbol{x}^\star)^T P_t(\boldsymbol{H})\boldsymbol{H}^2 P_t(\boldsymbol{H})(\boldsymbol{x}_0 - \boldsymbol{x}^\star))] \tag{27}$$

$$= \mathbb{E}_{\boldsymbol{H}}[(\lambda^2 P_t)(tr(H))^2] \tag{28}$$

$\square$

**Proposition 2** ((Pedregosa & Scieur, 2020)). *Let $P_t^l$ be defined as*

$$P_t^l := \arg\min_{P_t(0)=1} \int P_t^2(\lambda)\lambda^l d\nu(\lambda). \tag{6}$$

*Then $(P_t^l)$ is the family of residual orthogonal polynomials w.r.t. to $\lambda^{l+1}d\nu$.*

*Proof.* We differentiate the expression for the metrics w.r.t. to the coefficients of the polynomials:

$$\frac{d}{da_k}\left(\int \lambda^l P_t^2(\lambda)d\mu(\lambda)\right) = \int \lambda^l \frac{d}{da_k}\left(\sum_{k=0}^t a_k\lambda^k P_t(\lambda)\right)d\mu(\lambda) =$$

$$= 2 \cdot \left(\int \lambda^{l+k} P_t(\lambda)d\mu(\lambda)\right) = 0$$

This means that $P_t(\lambda)$ is orthogonal to any polynomial of degree $t-1$ w.r.t to the intern product $\langle\cdot,\cdot\rangle_{\lambda^{l+1}d\mu}$

$\square$

## B  GCM AND LAGUERRE METHOD DERIVATION

We will first state two lemmas that allow us to construct the optimal polynomials. With them in hand the procedure is trivial.

**Lemma 7.** *Let* $(\tilde{P}_t)$ *be a family polynomials following*

$$\tilde{P}_t(\lambda) = (\alpha_t + \beta_t\lambda)\tilde{P}_{t-1})\lambda) + \gamma_t\tilde{P}_{t-2}(\lambda),$$

*with* $\tilde{P}_0$ *a constant polynomial and* $\tilde{P}_t \neq 0, \forall t$. *Then*

$$P_t(\lambda) = (a_t + b_t\lambda)P_{t-1}(\lambda) + (1 - a_t)P_{t-2}(\lambda) \tag{29}$$

*is the recurrence for* $P_t(\lambda) = \tilde{P}_t(\lambda)/\tilde{P}_t(0)$. *With:*

$$a_t = \delta_t\alpha_t \tag{30}$$
$$b_t = \delta_t\beta_t \tag{31}$$
$$\delta_t = (\alpha_t + \gamma_t\delta_{t-1}) \quad (\delta_0 = 0) \tag{32}$$

The proof of this is presented in Pedregosa & Scieur (2020). Further, we know how to compute the recurrence for the polynomials of a shifted distribution:

**Lemma 8.** *Let* $(\tilde{P}_t)$ *be a family polynomials orthogonal w.r.t following*

$$\tilde{P}_t(\lambda) = (\alpha_t + \beta_t\lambda)\tilde{P}_{t-1}(\lambda) + \gamma_t\tilde{P}_{t-2}(\lambda), \tag{33}$$

*and define polynomials* $P_t$ *s.t. :*

$$P_t(m(\lambda)) = \tilde{P}_t(\lambda),$$

*with* $m(\lambda) = a\lambda + b$ *a non singular affine transform. Then* $P_t$ *follows a recurrence like in eq.* (33), *with:*

$$\alpha'_t = \alpha_t + b\beta_t \tag{34}$$
$$\beta'_t = a\beta_t \tag{35}$$
$$\gamma'_t = \gamma_t \tag{36}$$

The lemma is self-evident by considering eq. (33) with argument $m^{-1}(\lambda)$

These results are enough to get the recurrence relation for the residual polynomial w.r.t $x^\beta(L - x)^\alpha$. We begin by the standard Jacobi polynomials, which are orthogonal w.r.t $(1 - x)^\alpha(1 + x)^\beta$ and follow a recurrence according to $\alpha_t, \beta_t, \gamma_t$ below Szegö (1975):

$$\alpha_t = \frac{(2n + \alpha + \beta)(2n + \alpha + \beta - 1)}{2n(n + \alpha + \beta)} \tag{37}$$

$$\beta_t = \frac{(\alpha^2 - \beta^2)(2n + \alpha + \beta - 1)}{2n(n + \alpha + \beta)(2n + \alpha + \beta - 2)} \tag{38}$$

$$\gamma_t = \frac{-2(n + \alpha - 1)(n + \beta - 1)(2n + \alpha + \beta)}{2n(n + \alpha + \beta)(2n + \alpha + \beta - 2)} \tag{39}$$

We then shift the distribution according to $\eta(x)$, and then transform to the residual ones. We slightly simplify these computations and use remark 1 to get Algorithm 1.

We know (Szegö, 1975) that the Laguerre polynomials $L_t^\alpha$, with usual normalization, follow the recurrence

$$L_t^\alpha(\lambda) = \left(\frac{2t + \alpha - 1}{t} - \frac{1}{t}\lambda\right)L_{t-1}^\alpha(\lambda) + \frac{t + \alpha - 1}{t}L_{t-2}^\alpha(\lambda) \tag{40}$$

As we don't have to shift the domains, we have only to apply lemma 7 to get the Laguerre method.Further, we can get a explicit expression for $\delta_t = \frac{t}{t+\alpha}$, simplifying the expression.

---

**Algorithm 2:** Laguerre($\alpha$)

---

**Inputs**: Initial vector $\boldsymbol{x}_0$, function $f$
$\boldsymbol{x}_{-1} \leftarrow \boldsymbol{0}$
**for** $t = 1, \ldots, T$ **do**
$\quad \left\lfloor \quad \boldsymbol{x}_t \leftarrow \boldsymbol{x}_{t-1} + \frac{t-1}{t+\alpha}(\boldsymbol{x}_{t-1} - \boldsymbol{x}_{t-2}) - \frac{1}{t+\alpha}\nabla f(\boldsymbol{x}_{t-1}) \right.$

---

## C  PROOFS OF SECTION 3

In the following we will consider shifted versions of the spectral distributions. This shift is written as an affine transform $m(\lambda) : [0, L] \to [-1, 1]$ because most results in the theory of orthogonal polynomials are stated in terms of distributions supported in $[-1, 1]$.
This can be seen as an additional layer of abstraction because the quantities evaluated with the shifted distributions and polynomials are proportional, i.e. if $P_t(x) = \tilde{P}_t(m(x))$ and $\mu'(x) = \tilde{\mu}'(m(x))$:

$$\int P_t^2(x)\mu'(x)\,\mathrm{d}x \propto \int \tilde{P}_t^2(x)\tilde{\mu}'(x)\,\mathrm{d}x \tag{41}$$

So all the asymptotics are the same. The Jacobi polynomials $P_t^{\alpha,\beta}$ are those orthogonal w.r.t $d\mu(x) = (1-x)^\alpha(1+x)^\beta$. Most works use the normalization $\tilde{P}_t^{\alpha,\beta}(-1) = (-1)^t \binom{t+\beta}{t}$. We will write $\tilde{P}_t^{\alpha,\beta}$ for this normalization and $P_t^{\alpha,\beta}$ for the residual polynomials

**Theorem 2** (GCM average-case rates). *A Generalized Chebyshev Method with parameters $(\alpha, \beta)$ applied to a problem with e.s.d. as in Assumption 1 has average-case rates*

$$\mathbb{E}[f(\boldsymbol{x}_t) - f(\boldsymbol{x}^\star)] \sim L \cdot C_{1,\nu}^{\alpha,\beta} \begin{cases} t^{-1-2\beta} & \text{if } \alpha < \tau + 1/2 \text{ and } \beta < \xi + 3/2 \\ t^{-2(\xi+2)}\log t & \text{if } \alpha = \tau + 1/2 \text{ and } \beta = \xi + 3/2 \\ t^{2(\max\{\alpha-\beta-\tau,-\xi-1\}-1)} & \text{if } \alpha > \tau + 1/2 \text{ or } \beta > \xi + 3/2 \end{cases}, \tag{13}$$

$$\mathbb{E}[\|\nabla f(\boldsymbol{x}_t)\|_2^2] \sim L^2 \cdot C_{2,\nu}^{\alpha,\beta} \begin{cases} t^{-1-2\beta} & \text{if } \alpha < \tau + 1/2 \text{ and } \beta < \xi + 5/2 \\ t^{-2(\xi+3)}\log t & \text{if } \alpha = \tau + 1/2 \text{ and } \beta = \xi + 5/2 \\ t^{2(\max\{\alpha-\beta-\tau,-\xi-2\}-1)} & \text{if } \alpha > \tau + 1/2 \text{ or } \beta > \xi + 5/2 \end{cases}, \tag{14}$$

*where $C_\nu^{\alpha,\beta}$ is a distribution dependent constant.*

*Proof.* We will prove that for any $\alpha$ and $\beta$, $\xi, \tau > -1$, $l > 0$ and $\nu$ following Assumption 1, we have

$$\int P_t^{\alpha,\beta}(x)^2 x^l d\nu_{\tau,\xi-l}(x) \sim L^l C_\nu^{\alpha,\beta} \begin{cases} t^{-1-2\beta} & \text{if } \alpha < \tau + 1/2 \text{ and } \beta < \xi + 1/2 \\ t^{-2(\xi+1)}\log t & \text{if } \alpha = \tau + 1/2 \text{ and } \beta = \xi + 1/2 \\ t^{2(\max\{\alpha-\beta-\tau,-\xi\}-1)} & \text{if } \alpha > \tau + 1/2 \text{ or } \beta > \xi + 1/2 \end{cases}$$

We will first show this result for the Beta weights, then show that distributions with the same concentration behave similarly.
The normalization of $\tilde{P}_t^{\alpha,\beta}$ is s.t.[ Szegö (1975) (4.3.3)]:

$$\int_{-1}^1 \tilde{P}_t^{\alpha,\beta}(x)(1-x)^\alpha(1+x)^\beta\,\mathrm{d}x = \frac{2^{\alpha+\beta+1}}{2n+\alpha+\beta+1}\frac{\Gamma(n+\alpha+1)\Gamma(n+\beta+1)}{\Gamma(n+1)\Gamma(n+\alpha+\beta+1)} = \Theta(t^{-1}) \tag{42}$$

Further, the residual polynomials are s.t. $|P_t^{\alpha,\beta}| = \Theta(t^{-\beta})|\tilde{P}_t^{\alpha,\beta}|$, from the definition of the classical normalization.
We state the result (Exercise 91, Generalisation of 7.34.1) from Szegö (1975):

**Lemma 9.** *We have*

$$\int_0^1 (1-x)^\tau P_t^{\alpha,\beta}(x)^2 dx \sim \Theta(h_\tau^\alpha) \tag{43}$$

$$h_\tau^\alpha := \begin{cases} t^{2(\alpha-\tau-1)} & \text{if } \alpha > \tau + 1/2 \\ t^{-1}\log t & \text{if } \alpha = \tau + 1/2 \\ t^{-1} & \text{if } \alpha < \tau + 1/2 \end{cases} \tag{44}$$

Noting that $\tilde{P}_t^{\alpha,\beta}(x) = (-1)^t \tilde{P}_t^{\beta,\alpha}(-x)$, we can write:

$$\int_{-1}^1 \tilde{P}_t(x)^2 (1-x)^\tau (1+x)^\xi dx = \Theta\left(\int_0^1 (1-x)^\tau |\tilde{P}_t^{\alpha,\beta}(x)|^2 dx\right) + \Theta\left(\int_0^1 (1-x)^\xi |\tilde{P}_t^{\beta,\alpha}(x)|^2 dx\right) \tag{45}$$

We can then show our result for $d\nu_{\tau,\xi-l}(x) = x^{\xi-l}(L-x)^\alpha$ by carefully considering each of the cases on Lemma 9 and the maximum of each term in eq. 45, and an added $t^{-2\beta}$ from the different normalization. With this, we have the wanted result for the Beta weights
It remains to show:

$$\int_0^1 \tilde{P}_t^{\alpha,\beta}(x)^2 \, d\nu_{\tau,\xi}(x) = \Theta\left(\int_0^1 (1-x)^\tau \tilde{P}_t^{\alpha,\beta}(x)^2 dx\right) \tag{46}$$

And the rest follows from the same arguments. We do this with the help of this lemma shown in Van Assche (1995) relating to the weak convergence of the orthogonal polynomials:

**Lemma 10.** *Let $\mu$ be a measure and $(p_t)$ it's family of orthonormal polynomials s.t. $p_t$ follow the recurrence:*

$$xp_t(x) = a_t p_{t+1}(x) + b_t p_t(x) + a_{t-1} p_{t-1}(x)$$

*and $a_t, b_t$ converge respectively to $a, b$. Then for any $f$ continuous and bounded:*

$$\int f(x) p_t^2(x) d\mu(x) \to \frac{1}{\pi} \int_{-1}^1 \frac{f(x)}{\sqrt{1-x^2}} dx \tag{47}$$

Let $\epsilon$ s.t.

$$x \geq 1 - \epsilon \Rightarrow |\,d\nu_{\tau,\xi} - A(1-x)^\tau| \leq B(1-x)^\tau \tag{48}$$

We observe that for $0 < x < 1 - \epsilon$, $f(x) = \frac{d\nu_{\tau,\xi}}{(1-x)^\alpha(1+x)^\beta}$ is bounded.
We get from an application of 10, and the observation that $\tilde{P}_t^{\alpha,\beta} = \mathcal{N}_t p_t^{\alpha,\beta}$, with $\mathcal{N}_t = \Theta(t^{-1/2})$:

$$\underbrace{\int_0^1 (1-x)^\tau \tilde{P}_t^{\alpha,\beta}(x)^2 \, dx}_{\Theta(h_\tau^\alpha)} = \underbrace{\int_0^{1-\epsilon} (1-x)^\tau \tilde{P}_t^{\alpha,\beta}(x)^2 \, dx}_{\Theta(t^{-1})} + \int_{1-\epsilon}^1 (1-x)^\tau \tilde{P}_t^{\alpha,\beta}(x)^2 \, dx \Rightarrow \tag{49}$$

$$\int_{1-\epsilon}^1 (1-x)^\tau \tilde{P}_t^{\alpha,\beta}(x)^2 \, dx = \Theta(h_\tau^\alpha) \tag{50}$$

$$\int_0^1 \tilde{P}_t^{\alpha,\beta}(x)^2 \, d\nu_{\tau,\xi}(x) = \underbrace{\int_0^{1-\epsilon} \tilde{P}_t^{\alpha,\beta}(x)^2 f(x)(1-x)^\alpha(1+x)^\beta \, dx)}_{\Theta(t^{-1})}$$

$$+ \Theta\left(\underbrace{\int_{1-\epsilon}^1 (1-x)^\tau \tilde{P}_t^{\alpha,\beta}(x)^2 \, dx}_{\Theta(h_\tau^\alpha)}\right) \tag{51}$$

$\square$

**Theorem 3** (Optimal Rates). *Let $\nu$ follow Assumption 1. The optimal asymptotic average-case rates for $\mathbb{E}[f(\boldsymbol{x}_t) - f(\boldsymbol{x}^\star)]$ and $\mathbb{E}[||\nabla f(\boldsymbol{x}_t)||_2^2]$ are attained by the GCM with parameters $(\tau, \xi + 2)$ and $(\tau, \xi + 3)$, respectively, and read*

$$\mathbb{E}[f(\boldsymbol{x}_t) - f(\boldsymbol{x}^\star)] = \Theta(t^{-2(\xi+2)}), \qquad \mathbb{E}[||\nabla f(\boldsymbol{x}_t)||_2^2] = \Theta(t^{-2(\xi+3)}).$$

*Proof.* We will prove that for $\tau, \xi > -1$ If $\alpha = \tau$ and $\beta = \xi + l + 1$ (i.e., are optimal), the rate of convergence reads

$$\min_{P_t(0)=1} \int P_t^2(\lambda)\lambda^l d\nu(\lambda) = \Theta\left(\int_0^l \tilde{P}_t^{\alpha,\beta}(\lambda)^2 (L-\lambda)^\tau \lambda^{\xi+l}\, \mathrm{d}\lambda\right) = \Theta(t^{-2(\xi+l+1)}) \qquad (52)$$

Showing the second equality is easy by considering theorem 2, and that is further the minimum asymptotic rate for the Beta distribution $\mu_{\tau,\xi}$.

By setting $p_t^\nu$ and $P_t^\nu = \frac{p_t^\nu}{p_t^\nu(0)}$ the optimal orthonormal and residual and polynomials w.r.t. $\nu$ we show that $P_t^\nu$ must have the same rate on $\mu_{\tau,\xi}$ as it does on $\nu$, thus the optimal rate of $\nu$ cannot be lower than the optimal rate of $\mu_{\tau,\xi}$. Indeed, setting $\epsilon_1, \epsilon_2$ as in eq. 48:

$$\int_{1-\epsilon_2}^1 P_t^\nu(x)^2 d\nu(x) = \left(\Theta \int_{1-\epsilon_2}^1 P_t^\nu(x)^2 d\mu_{\tau,\xi}(x)\right) \qquad (53)$$

$$\int_{-1}^{-1+\epsilon_1} P_t^\nu(x)^2 d\nu(x) = \Theta\left(\int_{-1}^{-1+\epsilon_1} P_t^\nu(x)^2 d\mu_{\tau,\xi}(x)\right) \qquad (54)$$

$$\int_{-1+\epsilon_1}^{1-\epsilon_2} P_t^\nu(x)^2 d\nu(x) = \Theta\left(\int_{-1+\epsilon_1}^{1-\epsilon_2} P_t^\nu(x)^2 d\mu_{\tau,\xi}(x)\right) = \Theta\left(\frac{1}{p_t^\nu(-1)^2}\right) \qquad (55)$$

$$\qquad (56)$$

Where the first two equations come from the fact that $\nu = \Theta(\mu_{\tau,\xi})$ near $-1$ and $1$ and the third from lemma 10.

This effectively upper bounds the rates on $\nu$ because the rates of $P_t^\nu$ on $\mu_{\tau,\xi}$ can't be lower than $-2(\xi + l + 1)$. $\qquad \square$

**Proposition 3** (GCM worst-case rates). *Let $f$ be a convex, L-smooth quadratic function. Then, for the Generalized Chebyshev Method with parameters $(\alpha, \beta)$, we have worst-case rates*

$$f(\boldsymbol{x}_t) - f(\boldsymbol{x}^\star) \leq C_1 L \begin{cases} t^{2(\alpha-\beta)} & if \quad \alpha > \beta - 1 \\ t^{-1-2\beta}, & if \quad \alpha \leq \beta - 1 \quad \beta \leq \frac{1}{2} \\ t^{-2}, & if \quad \alpha \leq \beta - 1 \quad \beta \geq \frac{1}{2} \end{cases}, \qquad (15)$$

$$||\nabla f(\boldsymbol{x}_t) - f(\boldsymbol{x}^\star)|| \leq C_2 L^2 \begin{cases} t^{2(\alpha-\beta)} & if \quad \alpha > \beta - 2 \\ t^{-1-2\beta}, & if \quad \alpha \leq \beta - 2 \quad \beta \leq 3/2 \\ t^{-4}, & if \quad \alpha \leq \beta - 2 \quad \beta \geq 3/2 \end{cases}. \qquad (16)$$

*Proof.* rates] We will prove that: $\sup_{x\in[0,L]} x^l P_t^{\alpha,\beta}(x)^2 = O(L^l t^{v(\alpha,\beta,l)})$. Where:

$$v(\alpha,\beta,l) = \begin{cases} 2(\alpha-\beta) & if \quad \alpha > \beta - l \\ -1-2\beta, & if \quad \alpha \leq \beta - l \quad \beta \leq l - \frac{1}{2} \\ -2l, & if \quad \alpha \leq \beta - l \quad \beta \geq l - \frac{1}{2} \end{cases} \qquad (57)$$

From Szegö (1975), Theorem 7.32.2, if $\theta < \frac{\pi}{2}$:

$$\tilde{P}_t^{\alpha,\beta}(\cos\theta) = \begin{cases} O(t^{-1/2}) & if \quad \alpha < -\frac{1}{2} \\ O(t^\alpha) & if \quad \alpha \geq -\frac{1}{2}, 0 \leq \theta \leq ct^{-1} \\ \theta^{-\alpha-1/2}O(t^{-1/2}) & if \quad \alpha \geq -\frac{1}{2}, \theta > ct^{-1} \end{cases} \qquad (58)$$

We observe that, from the symmetry of the Jacobi polynomials:

$$\sup_{x\in[0,L]} x^l P_t^{\alpha,\beta}(x)^2 = \Theta\left(\max\left\{\sup_{x\in[0,1]} P_t^{\alpha,\beta}(x)^2, \sup_{x\in[0,1]} (1-x)^l P_t^{\beta,\alpha}(x)^2\right\}\right) \qquad (59)$$

The $(1-x)^l$ term, corresponds to $(2\sin(\frac{\theta}{2}))^l$ in the variable $\theta$, which is $O(\theta^{2l})$. The rest follows from carefully considering the expressions given by eq. 58. $\qquad \square$

**Theorem 4** (Nesterov average-case rates). *Let $\nu$ as in Assumption 1. Then for the Nesterov method, we have average-case rates*

$$\mathbb{E}[f(\boldsymbol{x}_t) - f(\boldsymbol{x}^\star)] \sim C'_{1,\nu} \left\{ \begin{array}{ll} t^{-2(\xi+2)} & if\ \xi < -1/2 \\ t^{-3} \log t & if\ \xi = -1/2 \\ t^{-(\xi+7/2)} & if\ \xi > -1/2 \end{array} \right. , \quad \mathbb{E}[\|\nabla f(\boldsymbol{x}_t)\|_2^2] \sim C'_{2,\nu} t^{-(\xi+9/2)}. \quad (17)$$

*Proof.* We will prove:

$$\int_0^1 P_t^{\text{Nes}}(\lambda)^2 \lambda^l \, d\nu_{\tau,\xi-l} \sim C'_\nu \left\{ \begin{array}{ll} t^{-2(\xi+1)} & \text{if } 0 < \xi < 1/2 \\ t^{-3} \log t & \text{if } \xi = 1/2 \\ t^{-(\xi+5/2)} & \text{if } \xi > 1/2 \end{array} \right. \quad (60)$$

Paquette et al. (2020) has shown that the nesterov polynomials $P_t$ are asymptotically, in $t$:

$$P_t(\lambda) \sim \frac{2J_1(t\sqrt{\alpha\lambda})}{t\sqrt{\alpha\lambda}} e^{-\alpha\lambda t/2} \quad (61)$$

In the sense that:

$$\int_0^1 u^l \left[ \widetilde{P_t^2}(u) - \frac{4J_1^2(t\sqrt{u})}{t^2 u} e^{-ut} \right] 4 \, d\mu_{MP}(u) = O(t^{-(l+25/12)}) \quad (62)$$

The arguments can be easily used to show that such an integral is $O(t^{-(\alpha+l+31/12)})$ when evaluated wrt a general $d\mu$ s.t $\mu' = \Theta(\lambda^\alpha)$ near 0.

We can thus consider our integral of interest substituting $P_t^{\text{Nes}}$ by it's Bessel asymptotic and dividing it into three regions, i.e. $[0,1] = [0, \frac{\epsilon}{t}] \cup [\frac{\epsilon}{t}, \frac{\epsilon}{\sqrt{t}}] \cup [\frac{\epsilon}{\sqrt{t}}, 1]$ corresponding to two different regimes for the Bessel function. The first region will give us the asymptotic and the others we will bound.

We consider first, for some $\epsilon > 0$:

$$\int_{\frac{\epsilon}{t}}^{\frac{\epsilon}{\sqrt{t}}} u^\xi \frac{4J_1^2(t\sqrt{u})}{t^2 u} e^{-ut} \, du \quad (63)$$

We note the asymptotic for $J_1^2$:

$$J_1^2(\sqrt{tv}) \sim \frac{1}{\pi\sqrt{tv}} (1 + \cos(2\sqrt{tv} + 2\gamma)) \quad (64)$$

Doing the change of variable $v = tu$, and identifying the upper limit of the interval, which is $\epsilon t^{1/2}$ to $\infty$:

$$\int_{\frac{\epsilon}{t}}^{\frac{\epsilon}{\sqrt{t}}} u^\xi \frac{4J_1^2(t\sqrt{u})}{t^2 u} e^{-ut} \, du = \Theta\left( t^{-2-\xi} \int_\epsilon^\infty v^{\xi-1} J_1^2(\sqrt{tv}) e^{-v} \, dv \right) \quad (65)$$

$$= \Theta\left( t^{-2-\xi} \int_\epsilon^\infty v^{\xi-1} \frac{1}{\pi\sqrt{tv}} e^{-v} \, dv \right) \quad (66)$$

$$= \Theta\left( t^{-\frac{5}{2}-\xi} \underbrace{\int_\epsilon^\infty v^{\xi-\frac{3}{2}} \frac{1}{\pi\sqrt{tv}} e^{-v} \, dv}_{\Gamma(\xi-\frac{1}{2},\epsilon)} \right) \quad (67)$$

Where the cosinus term goes to 0 from the Riemann-Lebesgue lemma and $\Gamma$ is the incomplete Gamma function.

The term corresponding to the interval $[\epsilon t^{-1/2}, 1]$ is exponentially small. Indeed, because of the exponential $e^{-ut}$ it is $O(e^{-\epsilon\sqrt{t}})$. This shows that the integral concentrates in a region that is closer and closer to 0 and that only the behaviour of the distribution near 0 matters. We have for the $[0, \frac{\epsilon}{t}]$ region, doing the change of variables $v = t^2 u$:

$$\int_0^{\frac{\epsilon}{t}} u^\xi \frac{4J_1^2(t\sqrt{u})}{t^2 u} e^{-ut} \, du = \Theta\left( t^{-2(\xi+1)} \int_0^{t\epsilon} v^\xi \frac{J_1^2(\sqrt{v})}{v} e^{-\frac{v}{t}} \, dv \right) \quad (68)$$

And the $e^{\frac{-v}{t}}$ is $\Theta(1)$. We have the following Bessel asymptotics:

$$\frac{J_1^2(\sqrt{v})}{v} \sim \frac{1}{4}, \qquad\qquad v \to 0 \tag{69}$$

$$\frac{J_1^2(\sqrt{v})}{v} = O(v^{-3/2}), \qquad v \to \infty \tag{70}$$

So we divide this integral aswell:

$$t^{-2(\xi+1)} \int_1^{t\epsilon} v^\xi \frac{J_1^2(\sqrt{v})}{v} e^{-\frac{v}{t}} \, dv = \Theta\left(t^{-2(\xi+1)} \int_\epsilon^{t\epsilon} v^{\xi-\frac{3}{2}} \, dv\right) = \Theta\left(I_\xi(t) t^{-\xi-\frac{5}{2}}\right) \tag{71}$$

$$t^{-2(\xi+1)} \int_0^1 v^\xi \frac{J_1^2(\sqrt{v})}{v} e^{-\frac{v}{t}} \, dv = \Theta\left(t^{-2(\xi+1)} \int_0^\epsilon \epsilon^1 v^\xi \, dv\right) = \Theta\left(t^{-2(\xi+1)}\right) \tag{72}$$

Where $I_\xi(t) = \log t$ if $\xi = \frac{1}{2}$ and 1 otherwise.
The nesterov rate is then $I_\xi(t) t^{-\xi-\frac{5}{2}}$ if $\xi \geq \frac{1}{2}$ and $t^{-2(\xi+1)}$ if $0 < \xi < \frac{1}{2}$ $\qquad\square$

**Theorem 5** (Gradient Descent average-case rates). *Let $\nu$ as in Assumption 1. Then for gradient descent*

$$\mathbb{E}[f(\boldsymbol{x}_t) - f(\boldsymbol{x}^\star)] = \Theta(t^{-(\xi+2)}), \qquad \mathbb{E}[||\nabla f(\boldsymbol{x}_t)||_2^2] = \Theta(t^{-(\xi+3)}). \tag{18}$$

*Proof.* Considering that $P_t^{\text{GD}}(\lambda) = (1 - \frac{\lambda}{L})^t$ we will prove :

$$\int_0^1 (1-\lambda)^{2t} \lambda^l \, d\nu_{\tau,\xi-l} = \Theta(t^{-(\xi+l+1)} \tag{73}$$

We know, for the Beta weights, that:

$$\int_0^1 (1-\lambda)^{2t+\tau} \lambda^{\xi+l} \, d\lambda = \frac{\Gamma(l+\xi+1)\Gamma(2t+\tau+1)}{\Gamma(2t+l+\xi+\tau+2)} = \Theta(t^{-(\xi+l+1)}) \tag{74}$$

We can indentify this asymptotic to the interval $\int_0^\epsilon$ for any $\epsilon$ because:

$$\int_\epsilon^1 (1-\lambda)^{2t+\tau} \lambda^{\xi+l} \, d\lambda = \mathcal{O}((1-\epsilon)^{2t}) \tag{75}$$

Then:

$$\int_\epsilon^1 (1-\lambda)^{2t} \lambda^l \, d\nu_{\tau,\xi-l} = \mathcal{O}((1-\epsilon)^{2t}) \tag{76}$$

$$\int 0^\epsilon (1-\lambda)^{2t} \lambda^l \, d\nu_{\tau,\xi-l} = \Theta\left(\int_0^\epsilon (1-\lambda)^{2t+\tau} \lambda^{\xi+l} \, d\lambda\right) = \Theta(t^{-(\xi+l+1)}) \tag{77}$$

$\qquad\square$

**Theorem 6** (Laguerre method rates). *Let $\alpha > -1$ and $\mu_\alpha$, $\alpha > -1$ be a Gamma distribution, i.e. $d\mu_\alpha(x) = \frac{x^\alpha e^{-x}}{\Gamma(\alpha+1)}$. The optimal rates are given by the Laguerre method of appropriate tuning and*

$$\mathbb{E}[f(\boldsymbol{x}_t) - f(\boldsymbol{x}^\star)] = \Theta(t^{-(\alpha+2)}). \tag{19}$$

*Proof.* Let $L_t^\alpha$ be the Laguerre polynomials with the usual normalization Szegö (1975):

$$\int L_t^\alpha(x)^2 d\mu_\alpha(x) = L_t^\alpha(0) = \binom{n+\alpha}{n} \tag{78}$$

We further now [Szegö (1975) (5.1.13)]]:

$$\sum_{k=0}^t L_t^\alpha(x) = L_t^{\alpha+1}(x) \tag{79}$$

Thus, letting $P_t^\alpha$ be the residual laguerre polynomial, we consider:

$$\mathbb{E}[f(\boldsymbol{x}_t) - f(\boldsymbol{x}^\star)] = \int P_t^{\alpha+2}(\lambda)^2 d\mu_{\alpha+1}(\lambda) = \binom{t+\alpha+2}{t}^{-2} \int L_t^{\alpha+2} d\mu_{\alpha+1}(\lambda)$$

$$= \binom{t+\alpha+2}{t}^{-2} \sum_{k=0}^{t} \left[ \int L_k^{\alpha+1}(\lambda) d\mu_{\alpha+1}(\lambda) \right]$$

$$= \binom{t+\alpha+2}{t}^{-2} \sum_{k=0}^{t} \binom{k+\alpha+1}{k} = \binom{t+\alpha+2}{t}^{-2} \binom{t+\alpha+2}{t}$$

$$= \binom{t+\alpha+2}{t}^{-1} = \Theta(t^{-(\alpha+2)})$$

(80)

□

## D    ADDITIONAL EXPERIMENTS

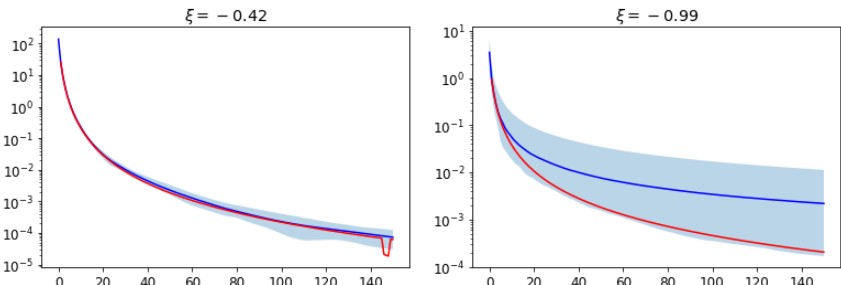

Figure 7: Empirical vs Theoretical function-value performance for $\text{GCM}(\alpha^\star, \beta^\star)$ . Red lines are given by numerical integration, shades are minimum and maximum values under 10 runs and the blue line is the mean

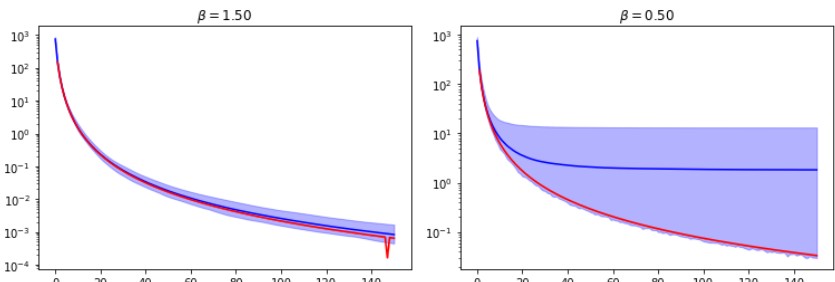

Figure 8: Empirical vs Theoretical function-value performance under Marchenko Pastur distribution. Red lines are given by numerical integration, shades are minimum and maximum values under 10 runs and the blue line is the mean

We note that in the regimes where the empirical average performance doesn't match the theoretical one, we can still find samples of problems who do match. This and the much larger variance on the function-value, this discrepancy is due to numerical unstability in these regimes.

