# OpenReview forum: "Only tails matter: Average-Case Universality and Robustness in the Convex Regime"
_ICLR.cc/2022/Conference — ICLR 2022 Submitted_

### Official Review · Reviewer_Tq5M · 2021-11-03

**Correctness:** 4
**Technical Novelty And Significance:** 2
**Empirical Novelty And Significance:** 2
**Recommendation:** 5
**Confidence:** 3

**Main Review:**

The paper seems to be a direct following paper of the two papers by Pedregosa and Scieur and the main technical difficulty is in the calculus part in the appendix. I don't consider this to be something very novel.

Minor problem: The definition of Marchenko Pastur distribution does not mention the point measure at 0. I guess it is because this point measure probably won't affect any rate derived in this paper because it gives an integral of 0. The authors can add some explanations to make this clear.

In Figure 6 and 7 in the appendix, there is no explanation for the blue line.

**Summary Of The Paper:**

This paper is a theoretical work concerning the convergence rate of random quadratic optimization problems. The authors provide a detailed analysis of the relation between convergence rate and ESD of the Hessian.

**Summary Of The Review:**

The paper is well-written and everything is explained clearly. But I don't think the paper contains enough technical and empirical novelty.

I consider this paper to be marginally below the acceptance threshold.

---

> ### Author Response · Authors · 2021-11-15
> **Addressing the issues**
>
> We address specific questions from the reviewer below:
> - **Lack of theoretical novelty**  We address this point in the common answer
> - **Lack of empirical novelty** We believe we have done illustrative experiments that show the validity of our theoretical results. We would like to ask the reviewer which kind of new experiments can be relevant for improving the manuscript.
> - **Why we do not consider a mass of 0.** Your observation about the mass at $0$ is correct. We added a comment on this right after Assumption 1.
> - **Lack of explanation for blue lines in Figures 6 and 7.** An explanation for  the blue lines on figures 5 and 7 has been added

---

> ### Author Response · Authors · 2021-11-26
> **Kind Reminder**
>
> Dear Reviewer,
> We thank you for the time and effort spent reviewing our paper. We wish to inform you that we have updated our manuscript as well as responded to your main concerns in our rebuttal below. We have also written a general response to all reviewers where we highlight the ways our work departs from previous literature and how our contributions add to the understanding of the optimization dynamics of first-order methods. We hope our responses were satisfactory in clarifying any remaining points that hinder you from potentially reconsidering your evaluation of our work. We are also happy to respond to any new guidance or questions that you may have

---

### Official Review · Reviewer_poH2 · 2021-11-04

**Correctness:** 4
**Technical Novelty And Significance:** 2
**Empirical Novelty And Significance:** 3
**Recommendation:** 5
**Confidence:** 4

**Main Review:**

The main strength of the paper is to go beyond Pedregosa/Scieur and consider the case where the limiting spectral measure is not assumed fully known. I agree with the authors that tail-type assumptions like the ones they make are more reasonable in practice. Moreover, they provide results that relate the parameters used to define the method to the convergence rates, and allow for comparisons.

There are three main weaknesses to the paper. One is that there is no significant new theory developed; all results essentially follow the same proof ideas as Pedregosa/Scieur. The second is that the paper is not at all impressive in terms of experiments, which would have helped because the theory is not that surprising. The third weakness is that the writing is unclear at at least two important points (1 and 2 below). The ideas in those places seem incorrect or contradictory. I'd still like to see these points better explained.

-- Major issues --

(1) It is nice that the authors give a full analysis of their "generalized Chebyshev method" under their e.s.d. assumption. However, I would have thought that the rate-optimal way to choose $\alpha,\beta$ would be the one given in the paragraph starting with "The optimal method w.r.t...." in page 5. This would align with the fact that orthogonal polynomials are optimal. However, it does not seem to be the case. What did I not understand?

(2) It is never true that a discrete spectral distribution equals a Beta, a Laguerre law, or satisfies Assumption 1. It seems to me that you need all of these to work asymptotically, but this is not mentioned in the paper. Given this, I am not sure of how exactly the proofs should be carried over: there may be difficulties with the recursion, maybe?

-- Minor issues --

The use of QuickSort in the introduction as an example of average-case analysis is a bit unfortunate, as it is completely unrelated to the paper at hand. Perhaps there's no need to give an example at that point.

Page 5, " "The optimal method w.r.t...." -- why do you speak of a metric $l$?

Page 5 again: can you give examples of random matrices with beta-shaped spectral distribution?

Appendix, section B: some details/pointers on Jacobi polynomials and their recorrence would be welcome. Also, "jacobi" is not capitalized at least in one instance.



**Summary Of The Paper:**

-- EDIT: I have updated my scores in response to clarifications --

The problem of optimizing a convex quadratic function via first order methods is considered. This is a well-understood problem from the worst case point of view, and its complexity will depend on the largest and smallest eigenvalues of the associated Hessian matrix. However, a nice recent average-case analysis by Pedrogosa and Scieur gives the following result: if the spectral density of the Hessian converges to a "nice" probability measure (such as the Manchenko-Pastur law), then first-order methods that are tailored to this density may converge faster. In fact, such methods can be obtained from the three-term recurrence for the orthogonal polynomials of the measure.

The present manuscript follows up on the Pedregosa/Scieur, but departs from in two specific ways. Firstly, it considers the case where the limiting spectral law is a beta distribution, and derives the corresponding iteration. Secondly, the authors analyze the performance of this iteration under "mispecification": that is, they allow for more general spectral measures than the beta, and only specify how these measures behave near the extremes. In spite of this, they are able to derive bounds of the same order as in the beta case.

A few other theoretical results are presented: worst-case rates for the method derived from the beta distribution; an analysis of Nesterov's method under conditions on the tail of the spectral measure, and a result on Laguerre spectral measures. These results generalize other theorems obtained in the Pedregosa/Scieur paper. The theory is complemented by a small numerical study.



**Summary Of The Review:**

The contributions of the paper are probably correct. However, the mathematical contribution is too closely tied to previous work. Moreover, the experimental contribution is not significant. Finally, there are the writing/correctness issues mentioned above. When put together, these observations justify my low score for this paper.

---

> ### Author Response · Authors · 2021-11-15
> **Addressing the issues**
>
> We address  specific questions from the reviewer below:
> -  **Optimal way of choosing  $\alpha, \beta$**    The optimal way, w.r.t. objective $l$, to choose $\alpha$, $\beta$ given $\tau,\xi$ is indeed given in the paragraph you mentioned. Could the reviewer clarify why he said  “However, it does not seem to be the case”?  Perhaps the confusion from the use of ‘objective $l$’, which we clarify below. Substituting $l=1,2$ will give the same values of $\beta$ as in Theorem 5.
> Assumption 1 is restrictive There seems to be a misunderstanding here. As any convex smooth distribution must show a concentration coefficient for each of its edges, Assumption 1 covers this entire class of distributions.  As the reviewer mentioned ‘discrete distributions’  there may be confusion between the empirical spectral distribution (the distribution of each *instance* in the finite dimension case), and the expected spectral distribution (the *expectation* of the empirical one over a class of problems). The empirical spectral distribution is indeed discrete, but we can reasonably assume its expectation over a class to be continuous.
> - **Mention to Quick-Sort** We have removed the mention of the QuickSort algorithm as it seems it’s more confusing than clarifying.
> - **Mention of ‘metric’ $l$** 'Objective $l$'  subsumes the gradient norm, function-value, and distance to the optimum metrics. We use this terminology to refer to the three of them, and their associated optimal methods in a concise fashion.
> - **Example of a random matrix showing a beta-shaped spectrum** A recent work (https://arxiv.org/pdf/2005.01100.pdf) shows certain random matrices show a E.S.D. shaped as Beta distribution. This, however, is not important for our manuscript as the Beta distributions show up first as a generalization of the Marchenko Pastur distribution and second, from the proof p.o.v., as an element of the 'equivalence classes' $\nu_{\tau,\xi}$ for which we can compute the precise rates.
> - **Pointer about Jacobi polynomials recurrence**  We added a clarification on the Jacobi recurrence in Appendix B.

---

> > ### Comment · Reviewer_poH2 · 2021-11-24
> > **I was wrong on two points; some additional points.**
> >
> > I thank the authors for their clarifications, especially the ones entitled "Optimal way of choosing..." and "mention of 'metric'...". The notation is now clearer, and the authors are correct that I had the empirical spectral distribution (and not the ESD), so my "Major issue (2)" is not really that much of an issue.
> >
> > I still respectfully disagree from the authors in terms of novelty, while admitting this is somewhat subjective. I have upgraded my score do 4, in any case.

---

### Official Review · Reviewer_KTu2 · 2021-11-05

**Correctness:** 4
**Technical Novelty And Significance:** 3
**Empirical Novelty And Significance:** 3
**Recommendation:** 8
**Confidence:** 4

**Main Review:**

-- Strengths:
- The paper is very well written. The problem and the results are stated clearly.
- The average case convergence rate for a large class of quadratic problems is established for GCM and Nestrov method and further it is shown that GCM with appropriate parameters is optimal.

-- Weaknesses:
- The main concern I have is that the setting considered is rather limited. Being limited to quadratic problems, though encountered frequently in practice, is very restrictive.
- The paper only talks about convergence rate of expected loss, expected distance to the optimal point and so on. It would be interesting to see if something can be said about the variance of these quantities as well. Referring to the motivating example of the paper, the Quicksort algorithm, we know that the average case complexity is $O(n\log n)$ but we also know that the complexity concentrates around its mean, i.e. with high probability it terminates with a complexity close to the average. In the absence of such high-probability bounds, knowledge of the variance might be the next best thing and it could give us an idea of how close to the mean we expect these values to be. It would be interesting to consider this in the future.

-- Minor Comments/Typos:
- Even though it is clearly stated in the Contributions Section, when I read the paper it was not clear to me at first that Algorithm 1 is a contribution of the paper. It would be helpful to make the contributions clear throughout the text.
- How does definition 3 generalize the Marchenko-Pastur distribution? The MP has three terms whereas the Beta only has two. Please explain.
- The numbering of Theorems is not consistent. E.g. we have Theorem 2.1 or Proposition 2.1 but then later the numbering format changes to Theorem 3, etc.
- Page 1, Introduction, 3rd paragraph: infinite -> infinity.
- Page 2, paragraph before Section 1.2: $e^{\lambda}$ -> $e^{-\lambda}$
- Page 4, Remark 1: the abbreviation for first order method (F.O.M.) is not defined before being used.


**Summary Of The Paper:**

The paper considers the problem of average convergence rate of first order methods on a given ensemble of quadratic problems. The authors propose the Generalized Chebyshev Method (GCM) and show that it is optimal when the e.s.d. is beta distribution. They also show that so long as we know the behavior of e.s.d. near the edges of its support, GCM still achieves the optimal rate. The authors finally consider the Nestrov method and derive its asymptotic average-case convergence rate.

**Summary Of The Review:**

The paper analyzes average case convergence rate of first order methods on quadratic problems and establishes very interesting results for different e.s.d.'s and knowledge of the e.s.d. only around the edges of the support. The main concern is the limited application of the results as only quadratic problems are considered.

---

> ### Author Response · Authors · 2021-11-15
> **Addressing the issues**
>
> We thank reviewer KTu2 for bringing the typos to our attention and we have corrected the numbering format.
> We address specific questions from the reviewer below:
> - **The setting is limited.** We do agree with the reviewer that the quadratic setting is currently restrictive. However, we point to the reviewer that given the current state of the art in the study of average-case complexity analysis of optimization algorithms, a proper analysis outside the quadratic setting is indeed a major open problem that is currently out of the scope of our paper. Indeed, we think we do not have the right tools yet to perform such an analysis: defining a distribution over the larger class of non-quadratic problems, for instance the class of smooth and convex functions, is an interesting and hard question already mentioned in (Pedregosa and Scieur 2020). Nevertheless,  we expect our results to hold in the larger setting in a local sense, i.e., if we initialize close enough to the optimum for non-quadratics. Moreover, variants of our results can be used as lower bound for the class of smooth convex functions.
> - **Concentration results.** Paquette et al. (https://arxiv.org/pdf/2006.04299.pdf) show that under reasonable assumptions, which our setting satisfies, there is a tight concentration around the mean in high dimensions. We added a remark regarding this. Moreover, our experiments show this concentration.
> - **How is definition 3 a generalization of the MP measure.**  The beta-weights can be seen as a generalization of the non-strongly convex instance of the Marchenko Pastur measure. It contains a total of 3 parameters, including maximum eigenvalue, while the non-strongly convex MP distribution contains 1.
> - **Highlight Algorithm 1 as a Contribution.**  A brief comment has been added to page 5.
> - **F.O.M. abbreviation not introduced.** The abbreviation is now introduced on page 3.

---

### Official Review · Reviewer_VZRV · 2021-11-07

**Correctness:** 3
**Technical Novelty And Significance:** 2
**Empirical Novelty And Significance:** 2
**Recommendation:** 5
**Confidence:** 4

**Main Review:**

The paper is a follow-up work of the two articles by Paquette et al., 2020 and Pedregosa & Scieur, 2020.

I feel that the authors tend to oversell their work. For example, the authors claim to have "a complete analysis of average-case convergence in non-strongly convex problems." I think the results require restrictive conditions such as Assumption 1 and an asymptotical view.

In particular, I wonder why the authors claim to have "established that the asymptotic convergence of first-order methods on quadratic problems in the convex regime..." Is there a clear correspondence between conditions like Assumption 1 and the convex regimes?

The paper also stated that the methods and their guarantees are more practical. I feel that asymptotic guarantees are rarely helpful for high-dimensional problems in practice. Also, I wonder how the authors are going to estimate the characteristic values in Assumption 1?

Finally, I don't understand how the authors show that the rates presented in those theorems are optimal. For example, could the authors please explain to me why Theorem 4 gives optimal rates? Its proof in the appendix is relatively brief and presents a list of claims. The only justification for these claims, like equations 50, 51, 52, and 53, seem to be the phrase "we argue."

**Summary Of The Paper:**

The paper analyzes average convergence rates of random quadratic optimization problems. The authors attempt to characterize the rates via the expected spectral distribution (e.s.d) of the random matrix (objective's Hessian) and show that the proposed algorithms work well asyptotically.

**Summary Of The Review:**

Overall, I don't find the paper very exciting as a follow-up work of some recent publications. The results' proofs seem to be oversimplified. And the convergence rates are only in the asymptotic sense. The assumptions also make the paper a mismatch for several of the authors' optimality and universality claims.

---

> ### Author Response · Authors · 2021-11-15
> **Addressing the issues**
>
> We address specific concerns from the reviewer below:
>
> - **How Assumption 1 corresponds to the convex regime and why our paper does a complete analysis of this regime.** In contrast to (Universal Average-Case Optimality of Polyak Momentum, Scieur and Pedregosa 2020) which covers the strongly convex case, we address the setting where the left edge of the support of the spectrum starts at 0. Assumption 1 covers any smooth and non-strongly convex spectrum, because  any such distribution must show a concentration coefficient for each of its edges. Thus our analysis covers the entire convex regime.
> - **The asymptotic view is limited.** There's indeed a tradeoff between the granularity of the rates and the granularity of the classes of distribution we consider. Our main contribution is showing that an important aspect of the convergence, the asymptotic rate, is dependent only on the concentration around the edges and our experiments show that our claims are qualitatively valid on the non-asymptotic regime. To get results valid over each of the classes  $\nu_{\xi,\tau}$ we need to leave the 'precise' (non-asymptotic) rates for the asymptotic ones. We ask the reviewer what kind of non-asymptotic results would contribute to the manuscript?
> - **The ‘characteristic values are hard to estimate.** The 'characteristic values' are indeed hard to estimate in practice. This was our main motivation to introduce robustness w.r.t. The misspecification of the distribution.
> - **Proof of the optimality of the rates is confusing.** We clarified the proof that the rates we give are optimal. The main point of the proof is that the optimal polynomial w.r.t. to a $\nu$ following Assumption 1 must have the same rate on $\nu$ as it does on the corresponding Beta distribution. Eqs. 50, 51, and 52 work similarly to the proof of theorem 4.

---

> ### Author Response · Authors · 2021-11-26
> **Kind Reminder**
>
> Dear Reviewer,
> We thank you for the time and effort spent reviewing our paper, it has allowed us to significatively improve our manuscript. We wish to inform you that we have updated our manuscript as well as responded to your main concerns in our rebuttal below. We have also written a general response to all reviewers where we highlight the ways our work departs from previous literature and how our contributions add to the understanding of the optimization dynamics of first-order methods. We hope our responses were satisfactory in clarifying any remaining points that hinder you from potentially reconsidering your evaluation of our work. We are also happy to respond to any new guidance or questions that you may have.

---

### Author Response · Authors · 2021-11-16
**Regarding the novelty of the paper**

We respectfully disagree with the claims that our work is a direct extension of the papers by (Pedregosa and Scieur. 2020) with no technical difficulties (see, e.g., reviews Tq5m and poH2).

Despite the common starting point of tying the convergence of first-order methods to integrals and polynomials, we come to very different conclusions through very different techniques that we summarized in Section 1.1 in the paper. We recall the important points here:
- We show that for an average case analysis we only need to know the behavior of the density function around its edges, while Pedregosa and Scieur (2020) showed that we needed to know the location and length of the support for strongly convex problems.
- Moreover, we show that the rate of convergence can be improved significantly, while in (Scieur and Pedregosa, 2020) the rate remains essentially the same as the worst-case.
- In addition, we introduced the novel concept of robustness w.r.t. misspecification of the concentration coefficients. This is an important point that was not covered by the previous work in this topic, that required the introduction of new mathematical tools.

We believe our contributions represent a new level of granularity between the usual worst-case analysis and the one given in the original work. Despite the asymptotical view, w.r.t. the number of the iterations of the algorithms, our experiments show our conclusions are qualitatively valid in the non-asymptotic regime for real data.

This adds to the understanding of the optimization dynamics at an intuitive level and of the practical efficiency of usual algorithms for minimizing convex functions. We also believe we did significant technical extensions on the original proofs to show the universality of the convergence rates, including a creative use of Lemmas 9 and 10 and techniques from asymptotic analysis.

---

### Decision · Program_Chairs · 2022-01-20

**Decision:**

Reject

**Comment:**

This paper studies the average convergence rate for first order methods on random quadratic optimization problems. Specifically it is a follow-up to work of Pedregosa and Scieur. They study the expected spectral distribution (e.s.d.) of the objective's Hessian and show asymptotic guarantees that work under some assumptions. In comparison to Pedregosa and Scieur, the main takeaway is that you only need to know the distribution at the edges as opposed to the entire spectrum in order to get the same improved convergence. However some reviewers felt that the contributions were oversold, and for example that Assumption 1 is quite restrictive.